# Treatment of Rice Stubble with *Pleurotus ostreatus* and Urea Improves the Growth Performance in Slow-Growing Goats

**DOI:** 10.3390/ani11041053

**Published:** 2021-04-08

**Authors:** Thansamay Vorlaphim, Pramote Paengkoum, Rayudika Aprilia Patindra Purba, Chalermpon Yuangklang, Siwaporn Paengkoum, Jan Thomas Schonewille

**Affiliations:** 1Department of Livestock and Fisheries, Ministry of Agriculture and Forestry, Vientiane 01000, Laos; thansamay.vorlaphim@yahoo.com; 2School of Animal Technology and Innovation, Institute of Agricultural Technology, Suranaree University of Technology, Nakhon Ratchasima 30000, Thailand; 3Department of Agricultural Technology and Environment, Faculty of Sciences and Liberal Arts, Rajamangala University of Technology Isan, Nakhon Ratchasima 30000, Thailand; chalermpon.yu@rmuti.ac.th (C.Y.); J.T.Schonewille@uu.nl (J.T.S.); 4Programme in Agriculture, Faculty of Science and Technology, Nakhon Ratchasima Rajabhat University, Nakhon Ratchasima 30000, Thailand; siwaporn.p@nrru.ac.th; 5Department of Farm Animal Health, Faculty of Veterinary Medicine, Utrecht University, 3584 Utrecht, The Netherlands

**Keywords:** digestibility, goat, growth performance, rice stubble, urea, value-added product, white-rod fungi

## Abstract

**Simple Summary:**

Fungi treatment is well established as a promising approach to upgrade the nutritional value of lignocellulosic biomass. This potency of fungi treatment is, however, primarily based on in vitro experiments, and extrapolation to practice is currently hindered, owing to a dearth of studies addressing the practical relevance of fungal treatment of high-fiber feed, such as rice straw and rice stubble. These potential biomasses are rife in Southeast Asian countries, coinciding with increasing rice production; however, it remains a big challenge to utilize rice stubble as a potential feed for ruminants. Similar to rice straw, rice stubble is traditionally eliminated through controlled burning, which is harmful to the environment. The aim of this study was to convert rice stubble into a new animal feed capable of increasing environmental friendliness. Using urea, it is well known to modify the lignification or silicification of lignocellulosic biomass. However, it remains scanty in combination with fungi treatment. Therefore, we treated rice stubble with either urea or oyster fungus (*Pleurotus ostreatus*) or a combination of these two treatments and offered these treated rice stubbles to slow-growing goats with the objective to study their effect on feed intake, digestibility, and fermentation end-products.

**Abstract:**

The objective of this study was to evaluate the efficacy of the fungal treatment (*Pleurotus ostreatus*) of urea-treated rice stubble on growth performance in slow-growing goats. Eighteen crossbred Thai-native x Anglo-Nubian male goats (average body weight: 20.4 ± 2.0 kg) were randomly assigned to three experimental total mixed rations containing 35% rice stubble (RS) that were either untreated (URS), urea treated (UTRS), or treated with urea and fungi (UFTRS). URS and UTRS were cultivated and harvested from an aseptically fungal spawn, incubated at 25–30 °C for 25 days. Indicators of growth performance were monitored, and feces were collected quantitatively to assess nutrient digestibility, during a 12-week feeding trial. All goats remained healthy throughout the experiment. The goats fed UFTRS had a lower feed conversion ratio (kg feed/kg growth) compared to goats fed URS or UTRS. Compared to URS, dietary UFTRS increased the nutrient digestibility of slow-growing goats, such as organic matter (OM) (+8.5%), crude protein (CP) (+5.5%), neutral detergent fiber (NDF) (+39.2%), and acid detergent fiber (ADF) (+27.4%). Likewise, dietary UFTRS tended to increase rumen ammonia concentrations, but rumen pH and volatile fatty acids were not affected by UFTRS. In conclusion, the present study indicates that the fungal treatment of RS is an effective tool to improve the growth performance of slow-growing goats.

## 1. Introduction

Within Southeast Asian countries, Thailand is a large producer of rice, and thus, high amounts of biomass, in the form of rice straw and rice stubble, are available for the feeding of ruminants [1]. These agricultural coproducts provide a source of roughage typically relevant in times of feed scarcity such as the dry season [2]. Despite being poorly fermented and the slow disappearance from the rumen, rice straw can be used as a roughage source for ruminants, but processing of the rice straw may increase its nutritional value. For instance, calcium hydroxide or urea have been used for modifying either the lignification or silicification of rice straw, and such treatments result in increased dry matter intake and milk production [3]. These benefit values of rice straw utilization have been gradually gaining interest by researchers in enhancing feed value for ruminants [2]. In the case of rice straw scarcity, rice stubble can also be considered a roughage source. The nutritional value of rice stubble is, however, poor, mainly because of its high fiber content [4]. Therefore, the processing of rice stubble to increase its nutritional value can be considered opportune.

Of interest, few previous studies [5,6,7] indicated that the treatment of lignocellulosic biomass with white-rot fungi is one of the promising tools to modulate the nutritional value of highly lignified material. The potential of various white-rot fungi species, including *Pleurotus ostreatus*, to increase the nutritional value of tough lignified material is primarily based on in vitro studies [5,7,8], whereas in vivo studies are needed to demonstrate the efficacy of fungal treatment of lignocellulosic biomass under feeding conditions. However, the extrapolation of the current in vitro results to practice is difficult because there is currently a lack of studies addressing the practical relevance of fungal treatment of high-fiber feed such as rice stubble.

Therefore, the aim of the present study was to evaluate the efficacy of fungal treatment (*Pleurotus ostreatus*) of urea-treated rice stubble on the growth performance of slow-growing goats, and it was hypothesized that the treatment of rice stubble with *Pleurotus ostreatus* increases the efficiency of feed conversion to body weight gain in slow-growing goats. To date, the treatment of rice straw with urea is generally practiced to increase its digestibility [3,4]. This method is perceived as an easy, reproducible, and relatively cheap approach for the further potential to increase the digestibility of rice straw, where it also increases the N content of the feed. The latter is considered relevant because the nitrogen content of rice straw is generally less than 2% on a dry matter (DM) basis [9] and thus limits rumen fermentation [10,11]. As a consequence, it was considered opportune to evaluate the efficacy of *Pleurotus ostreatus* on growth performance using urea-treated rice stubble as a control. A dietary treatment containing untreated rice stubble was used as a negative control.

## 2. Materials and Methods

### 2.1. Ethical Statement

The research was carried out in accordance with regulations on animal experimentation and the Guidelines for the Use of Animals in Research as recommended by the National Research Council of Thailand (U1-02632-2559). The Animal Ethics Committee of Suranaree University of Technology issued a statement approving the experimental protocol (SUT 4/2558).

### 2.2. Preparation of Treated Rice Stubble

Rice stubble (RS) was collected and sampled from random spots at a paddy field near SUT Organic Farm, Nakhon Ratchasima, Thailand (14°52′48′′ N, 102°0′54′′ E at an elevation of 243 m above sea level). Rice material remaining in RS was gently removed by hand. RS then was mechanically chopped into particles 2–5 cm in length. Thereafter, ~300 kg of boiling water was added to each 100 kg of RS, and the moisturized RS was left overnight to enable the water to penetrate the inner structures of the RS and minimize contamination by unwanted fungal spores.

The RS intended for the urea-fungi-treated RS (UFTRS) treatment was then transferred into plastic bags and inoculated with a previously prepared spawn of *Pleurotus ostreatus*, as described by Vorlaphim et al. [12], and the plastic bags with inoculated RS were incubated at ambient temperature (25–30 °C) for 25 days. On Day 21 of the incubation period, urea was added (2.5%) to terminate the fungal activity. The RS intended for the urea-treated RS (UTRS) treatment was prepared almost similar to that of the UFTRS treatment; i.e., 2.0% instead of 2.5% urea was added to the moisturized rice stubble, and, for obvious reasons, the RS was not inoculated with *Pleurotus ostreatus*. Due to the small differences in the dietary proportions of cassava hay, we adjusted the amounts of urea so as to attain isonitrogenous rations. Furthermore, it was considered opportune to evaluate the efficacy of *Pleurotus ostreatus* on growth performance using urea-treated rice stubble as a control because it is common practice to treat rice straw with urea. Thus, the effects of fungi treatment were evaluated using UTRS as a control. For instance, the dietary treatment containing untreated RS (URS) was used as a negative control. As a consequence, the aforementioned difference in urea treatment was deemed necessary to attain iso-nitrogenous rations. The chemical composition of URS, UTRS, and UFTRS is shown in Table 1.

### 2.3. Animals, Treatments, and Experimental Design

Eighteen crossbred Thai-native × Anglo-Nubian male goats with an initial average body weight of 20.4 kg (SD 2.0 kg, *n* = 18) were used. These typically slow-growing goats were obtained from the university farm of Suranaree University of Technology, Thailand. The experiment had a parallel design and lasted for 12 weeks. The goats were randomly assigned to their isonitrogenous, experimental rations; i.e., a total mixed ration (TMR) consisting of 35% rice stubble (RS) that was either untreated RS (URS), urea-treated RS (UTRS), or urea-fungi-treated RS (UFTRS, Table 2).

The experiment was preceded by a 14-day adaptation period, during which the animals could become accustomed to the RS-containing rations. Animal health management was in accordance with our previous study [13]. The goats were housed in individual metabolic cages (length 2.2 m × width 1.3 m × height 2 m) and the experimental rations were offered ad libitum, twice daily at 0700 and 1700 h. Feed refusals were recorded each 24 h period to calculate daily feed intake. Fresh drinking water was available at all times. Each animal was weighed daily, from the adaptation period (Days 1–21) to the feeding trial (Days 22–84).

### 2.4. Sample Collection

Feed was sampled on the following days of the feeding trial, from Days 22 to 28, Days 50 to 56, and Days 78 to 84. The daily samples were divided into two parts; the first part was analyzed for DM, and the second part was kept and pooled at the end of each sampling period. The pooled feed samples were then dried for 72 h at 60 °C, subsequently ground (1 mm screen) by a cutting mill (Retsch SM 100 mill; Retsch Gmbh, Haan, Germany), and stored in a sealed jar at ambient temperature until analysis.

During the aforementioned periods, feces were collected quantitatively from each goat. Each 24 h collection of feces was weighed, and ~5% of each collection was dried for 72 h at 60 °C. The dried samples were stored in plastic bags at ambient temperature (25–30 °C) pending analysis.

Rumen fluid was collected (~500 mL) on the next day following the period of feces collection with the use of a stomach tube connected to a manual pump [14,15,16]. The rumen fluid samples were taken ~30 min before the morning meal and 2 and 4 h after the morning feeding. Immediately after collection, the pH of the rumen fluid samples was recorded, and the rumen fluid samples were filtered through four layers of cheesecloth. Then, 10 mL of a 50% H_2_SO_4_ solution was added to 100 mL of filtered rumen fluid, and the mixture was subsequently centrifuged at 16,000× *g* for 15 min. The supernatant was stored at −20 °C until the analysis of ammonia and volatile fatty acids.

### 2.5. Chemical Analysis

Prior to the chemical analysis of feces, the samples were pooled per goat for each collection period and subsequently ground (1 mm screen). Feed and feces were analyzed for the analysis of crude ash, ether extract (EE), and crude protein (CP) contents using the standard procedures, as described by the AOAC [17]. Crude protein was calculated as N × 6.25. Neutral detergent fiber (NDF), acid detergent fiber (ADF), and acid detergent lignin (ADL) were determined using the standard procedures method as described by van Soest et al. [18].

Hemicellulose was calculated as NDF minus ADR. Ammonia contents of rumen fluid were determined according to Bremner and Keeney [19], and volatile fatty acids were measured by means of HPLC (RF-10AXmugiL; Shimadzu; Kyoto, Japan), as described by Zinn and Owens [20] with minor modification in peak analyses as given by Purba and Paengkoum [21].

### 2.6. Calculation and Statistical Analysis

Total tract digestibility was calculated as follows: [Nutrient intake (g/day)—fecal excretion of corresponding nutrient (g/day)]/Nutrient intake (g/day) × 100%.

All data (e.g., indices of growth performance including nutrient intake and nutrient digestibility) were subjected to ANOVA using the MIXED procedure of SAS 9.4. The statistical model used was:Y*_ij_* = μ + τ*_i_* + ε*_ij_*(1)
where Y*_ij_* = response variable, μ = overall mean, τ*_i_* = effect of treatment (*i* = 3) and ε*_ij_* = residual error. The covariance structure was compound symmetry, which was selected in the Kolmogorov–Smirnov test of the mixed model of SAS. The LSMEANS statement estimates the treatment means adjusted for the effect of the covariate. The data obtained from the indices of growth performance were adjusted by the inclusion of the covariance mean using initial body weight as a covariable. Tukey’s HSD test was used to identify rations with different effects on the variable involved. Throughout, the level of statistical significance was pre-set at *p* < 0.05.

All data related to ruminal fermentation end-product observed after 0, 2, and 4 h of feedings were processed as a completely randomized design with repeated measures using the MIXED procedure of SAS 9.4. The statistical model used was:Y*_ijkl_* = µ + P*_i_* + Q*_j_* + S*_k_* (P) + (P × Q)*_ij_* + e*_ijkl_*(2)
where Y*_ijkl_* is the observation, μ is the overall mean, P*_i_* is the fixed effect of diet (*i* = 1 to 3 URS, UTRS, and UFTRS), Q*_j_* is the fixed effect of the sampling time (*j* = 1 to 3, postfeeding at 0, 2, and 4 h), S*_k_* is the random effect of the rumen fluid nested within the time observation (*k* = 1 to 3, Days 28, 56, and 54), (P × Q)*_ij_* is the interaction between diet and sampling time, and e*_ijkl_* is the residual error. Akaike’s information criterion of the mixed model of SAS was performed to obtain the compound symmetry of the covariance structure. Data were checked for normal distribution by the Kolmogorov–Smirnov test [22]. Statistical significance of the diet effect was tested against variance of rumen fluid nested within the diet according to repeated measures design [23]. Least-square means are reported, and significance was declared at *p* < 0.05. Differences among means of diets and time were calculated by Tukey’s HSD at *p* < 0.05 [23].

## 3. Results

### 3.1. Growth Performance

Initial body weight (BW) of the goats (Table 3) allocated to URS and UTRS were similar, but the initial BW of the goats allocated to UFTRS was lower (*p* < 0.05) compared to the goats treated with UTRS. The unbalanced initial BW of the goats was acknowledged into a covariate model to determine growth performance. Despite the significant difference in initial BW, the final BW of the goats was found to be similar (*p* > 0.05) after 12 weeks of feeding the experimental rations. Consequently, the highest BW gain and growth rate were found in the goats fed UFTRS, and both values were significantly greater (*p* < 0.05) than the corresponding values observed in the goats fed URS. Daily feed intakes were similar between URS and UTRS; however, the goats fed UFTRS consumed more feed (*p* < 0.05) than the goats fed URS. Nevertheless, the feed conversion ratio (Table 3) of the animals fed UFTRS was found to be lower (*p* < 0.05) compared to the goats fed URS, but UTRS did not affect the on-feed conversion ratio in the slow-growing goats (*p* > 0.05).

### 3.2. Macronutrient Intake and Digestibility

The intake of organic matter (OM) was similar between URS and UFTRS (*p* > 0.05), and the highest OM intake was observed when the goats were fed UTRS (*p* < 0.05). In contrast, the highest value of OM digestibility (*p* < 0.05) was found when the goats were fed UFTRS (Table 4). Crude protein (CP) intakes were mirrored by the DM intakes, and the highest value of CP digestibility was found when the goats were fed UFTRS. The intakes of neutral and acid detergent fiber (NDF and ADF, respectively) were significantly higher when the goats were fed UTRS, but respective intakes were similar (*p* > 0.05) to those when UFTRS was fed. The feeding of UFTRS, but not URS and UTRS, resulted in significantly higher values of NDF and ADF digestibility. Compared to URS, the respective NDF and ADF digestibility values were higher (*p* < 0.05) when UFTRS was fed to the goats. The intake of hemicellulose was lowest (*p* < 0.05) when goats were fed UFTRS, but its digestibility was almost two times greater (*p* < 0.05) compared to UTRS. The goats fed UTRS had the highest intakes (*p* < 0.05) of nonstructural carbohydrates (NSC), but NSC digestibility was only affected (*p* < 0.05) when UFTRS was fed (Table 4).

### 3.3. Ruminal Fermentation End-Product

Generally, there is no interaction between treatment (URS vs. UTRS vs. UFTRS) and a postfeeding effect (0, 2, and 4 h) for the ruminal fermentation end-product in rumen fluids (*p* > 0.05, Table 5). Rumen pH and total volatile fatty acids (VFAs) were similar (*p* ≥ 0.120) among the dietary treatments (URS vs. UTRS vs. UFTRS, Table 5). For the three dietary treatments, three measurement days, and the three time points/day combined, the overall mean values on rumen pH value and total VFA concentrations were found to be 6.69 (SD 0.04, *n* = 3) and 87.0 mmol/L (SD 4.12, *n* = 3), respectively. The proportion of acetic acid (% of total VFA) was similar (*p* ≥ 0.80) among the treatments both before (0 h) and 4 h after the morning feeding, i.e., 59.2% (SD 0.21, *n* = 3) and 59.3% (SD 0.61, *n* = 3). However, proportions of acetic acid 2 h after feeding tended to be greater when UFTRS was fed, and the values were 58.9, 61.3, and 66.1% of total VFA for URS, UTRS, and UFTRS, respectively. In contrast to acetic acid, the proportions of propionic and butyric acid were not influenced by dietary treatments. For the three dietary treatments, the three measurement days, and the three time points/day combined, the overall mean values of the proportions of propionic and butyric acid (% of total VFA) were found to be 28.1% (SD 0.86, *n* = 3) and 11.7% (SD 0.55, *n* = 3), respectively.

Rumen ammonia concentration was greater (*p* = 0.002, Table 5) when the goats were fed UFTRS. For the three measurement days and the three time points/day combined, the mean rumen ammonia values were found to be 15.5, 15.9, and 18.3 mg/100 dL after the feeding of URS, UTRS, and UFTRS, respectively.

## 4. Discussion

To the best of the author’s knowledge, the current data show for the first time that the fungal treatment of urea-treated rice stubble improves the growth rate and feed conversion efficiency in slow-growing goats. If rice stubble could be acknowledged as rice straw, the current observation of the growth rate of the goats fed URS is in line with the results reported by Wanapat et al. [3]. The current result of the improvement of the feed conversion efficiency in goats fed UFTRS is in line with the observation that the fungal treatment of rice stubble caused a significantly greater OM digestibility, irrespective of the urea treatment of rice stubble (Table 4). This notion is substantiated by the fact that, at least numerically, OM intake was lowest when the goats were fed UFTRS. It thus appears that the previous in vitro data reported on the fungi-induced improvement of OMD [5,7,8] can be extrapolated, at least qualitatively, to practical feeding conditions.

In the current study, UFTRS versus UTRS did not stimulate OM intake. Unfortunately, there are currently no reports available to substantiate this observation. The lack of effect of UFTRS on OM intake is, however, not easy to explain because it does not seem to be in line with the observed increases in apparent NDF and ADF digestibility when UFTRS was fed. Indeed, the dietary content of rumen-digestible NDF is generally believed to be the main determinant of rumen fill and therefore of DM intake in ruminants with unrestricted access to roughage-based rations [24]. It, thus, can be speculated that the disappearance rate of NDF from the rumen was not affected by the fungal treatment of UFTRS. This notion is in line with the current observation that the feeding of UFTRS did neither affect rumen pH nor total VFA concentrations. Thus, the current observations do not support the idea that rumen fermentation is stimulated by UFTRS. It is therefore hypothesized that the UFTRS-induced increase in the total tract digestibility of NDF is explained by an increased hindgut fermentation of structural carbohydrates. It is needless to say that the current study does not provide further clues to support this idea, and future studies are required to test this hypothesis.

In general, white-rod fungi are the only fungi capable of degrading lignin in their substructures [25], such as guaiacyl-, syringyl-, and *p*-hydroxyphenyl units [26]. In the process of lignin degradation, cellulose, and hemicellulose are liberated and subsequently attacked by fungal glycoside hydroxylases [25], thereby yielding substrate for microbes and thus potentially improving the nutritional value of lignocellulosic biomass. In the present study, *Pleurotus ostreatus* was used to treat RS. However, it should be taken into consideration that the cause-and-effect relationship between fungal treatment and improvement in the nutritional value of the lignocellulosic biomass is not straightforward due to a great variety in both the fungal species and chemical background of the available lignocellulosic biomass [6]. Clearly, the current results do not preclude the possibility that white-rode species and/or varieties other than *Pleurotus ostreatus* are at least equally effective in increasing apparent fecal digestibility. Indeed, it is well known that *Pleurotus eryngii* as well as certain *Ceriporiopsis subvermispora* varieties have great potential to effectively degrade lignocellulosic biomass [7].

In contrast to expectations [3,4], urea treatment of rice stubble did not significantly affect OM digestibility. There was, however, a numerical increase in OM digestibility of 5.3%. The observed increase in OM digestibility is in line with data reported by Zhang et al. [27], who reported an increase of 3.3% in OM digestibility in goats fed urea-treated, instead of untreated, rice straw. Despite the lower impact of urea treatment on OM digestibility, the result of OM digestibility, as reported by Zhang et al. [27], was found to be statistically significant. A possible reason for the different outcome of this study compared with that of Zhang et al.’s [27] might be a difference in the probability values of type I error (SEM values are almost identical, i.e., 1.05 and 1.02, respectively), likely caused by the number of animals used. Zhang et al. [27] used nine goats per treatment, whereas in our study, six goats were used.

The fungal treatment of UFTRS caused greater rumen ammonia concentrations, but the explanation for this observation is unfortunately not straightforward due to the fact that N intakes were ultimately somewhat greater when the goats were fed UFTRS. On the other hand, it may be suggested that the fungal treatment of rice stubble rendered more protein available for rumen fermentation [5,28]. This suggestion is corroborated by data obtained from in vitro research conducted by Nayan et al. [29] showing that the fungal treatment of wheat straw with two different strains of *Pleurotus eryngii* increased the NPN fraction in buffered rumen fluid under in vitro conditions. Moreover, the increase in NPN was associated with a decrease in cell-wall-bound proteins. Perhaps *Pleurotus ostreatus* is also able to liberate proteins bound to cell walls, thereby explaining, at least partly, the greater rumen ammonia concentrations after the feeding of UFTRS. The latter notion may also explain the observed greater apparent digestibility of CP when the goats were fed the fungi-treated rice stubble.

## 5. Conclusions

It is concluded that the fungal treatment of urea-treated rice stubble is a practical tool to improve the feed conversion ratio in slow-growing goats. The improved growth performance is caused by an improved total tract nutrient digestibility. Further studies are, however, warranted to substantiate the current findings with a larger sample size and to understand the in vivo cause-and-effect relationship between the fungal treatment of rice stubble and total tract digestibility of structural carbohydrates in particular.

## Figures and Tables

**Table 1 animals-11-01053-t001:** Chemical composition of untreated rice stubble (URS), urea-treated rice stubble (UTRS), and urea-fungi-treated rice stubble (UFTRS).

Item	URS	UTRS	UFTRS
Dry matter (g/kg fresh weight)	960	239	230
Organic matter (g/kg DM)	849	793	771
Crude protein (g/kg DM)	25	41	95
Ether extract (g/kg DM)	9	8	8
Neutral detergent fiber (g/kg DM)	779	768	659
Acid detergent fiber (g/kg DM)	580	597	591
Acid detergent lignin (g/kg DM)	49	40	35

DM: dry matter.

**Table 2 animals-11-01053-t002:** Ingredient and chemical composition of total mixed ration (TMR) diets consisting of untreated rice stubble (URS), urea-treated rice stubble (UTRS), and urea-fungi-treated rice stubble (UFTRS).

Item	TMR Diet		
	URS	UTRS	UFTRS
Ingredient, g/kg DM			
Constant components ^1^	150	150	150
Cassava hay	470	485	493
Untreated rice stubble	350	-	-
Urea-treated rice stubble	-	350	-
Urea-fungi-treated rice stubble	-	-	350
Urea	30	15	7
Chemical composition, g/kg DM			
Dry matter (g/kg fresh weight)	463	488	420
Organic matter (g/kg DM)	918	922	845
Crude protein (g/kg DM)	122	120	123
Ether extract (g/kg DM)	15	16	16
Neutral detergent fiber (g/kg DM)	354	354	325
Acid detergent fiber (g/kg DM)	244	254	241
Acid detergent lignin (g/kg DM)	17	3	03
Non-structural carbohydrates ^2^	427	432	382

^1^ The constant components consisted of the following ingredients (% as fed): rice bran, 5.0; soybean meal, 3.0; molasses, 5.0; salt, 0.4; sulfur, 0.2; di-calcium phosphate, 0.5; limestone, 0.2; premix, 0.7. ^2^ Calculated as organic matter—crude protein—ether extract—neutral detergent fiber. DM: dry matter.

**Table 3 animals-11-01053-t003:** Selected indices of growth performance of slow-growing goats after the feeding of total mixed ration (TMR) diets consisting of untreated rice stubble (URS), urea-treated rice stubble (UTRS), and urea-fungi-treated rice stubble (UFTRS).

Item	TMR Diet			SEM	*p* Value
	URS	UTRS	UFTRS		
Body weight (kg)					
Initial	20.5 ^a^	22.0 ^a^	18.9 ^b^	0.68	0.022
Final	23.2	25.0	23.9	0.74	0.226
Gain	2.7 ^b^	3.1 ^a,b^	5.0 ^a^	0.58	0.036
Growth rate (g/d)	35.6 ^b^	39.6 ^a,b^	64.9 ^a^	8.25	0.036
Feed intake (g DM/d)	618.5 ^b^	658.5 ^b^	687.4 ^a^	6.62	0.003
Feed conversion ratio (g/g) ^1^	17.4 ^a^	16.6 ^a^	10.6 ^b^	1.39	0.015

^a,b^ Means in the same row with a different superscript differ significantly (*p* < 0.05). ^1^ Calculated as g of feed/g of body weight gain. SEM: standard error of mean. DM: dry matter.

**Table 4 animals-11-01053-t004:** Mean macronutrient intake and apparent fecal digestibility after slow-growing goats fed total mixed ration (TMR) diets consisting of untreated rice stubble (URS), urea-treated rice stubble (UTRS), and urea-fungi-treated rice stubble (UFTRS). The values represent the overall means obtained during three distinct periods of the 12-week feeding trial.

Item	TMR Diet			SEM	*p* Value
	URS	UTRS	UFTRS		
Organic matter					
Intake (g/d)	567.9 ^b^	607.0 ^a^	580.8 ^a,b^	5.77	0.041
Digestibility (% of intake)	62.6 ^b^	65.9 ^b^	71.5 ^a^	1.02	0.010
Crude protein					
Intake (g/d)	75.5 ^b^	79.2 ^b^	84.7 ^a^	0.81	0.001
Digestibility (% of intake)	57.6 ^b^	63.6 ^ab^	67.1^a^	1.51	0.050
Neutral detergent fiber					
Intake (g/d)	218.9 ^b^	233.2 ^a^	223.0 ^a,b^	2.22	0.050
Digestibility (% of intake)	46.9 ^b^	53.6 ^a,b^	65.3 ^a^	2.31	0.017
Acid detergent fiber					
Intake (g/d)	150.9 ^b^	167.1 ^a^	165.3 ^a^	1.61	0.002
Digestibility (% of intake)	44.9 ^b^	51.1 ^a,b^	57.2 ^a^	2.05	0.018
Hemicellulose ^1^					
Intake (g/d)	67.9 ^a^	66.1 ^a^	57.7 ^b^	0.66	<0.001
Digestibility (% of intake)	35.1 ^b^	42.5 ^b^	69.1 ^a^	3.08	<0.001
Nonstructural carbohydrates ^2^					
Intake (g/d)	264.2 ^b^	284.1 ^a^	262.3 ^b^	2.64	<0.001
Digestibility (% of intake)	78.1 ^b^	80.6 ^b^	82.4 ^a^	0.23	0.020

^a,b^ Means in the same row with a different superscript, differ significantly (*p* < 0.05). ^1^ Calculated as neutral detergent fiber—acid detergent fiber. ^2^ Calculated as organic matter—crude protein—ether extract—neutral detergent fiber. SEM: standard error of mean.

**Table 5 animals-11-01053-t005:** A postfeeding effect (0 h, 2 h, and 4 h) for fermentation end-product in rumen fluids of slow-growing goats the feeding of total mixed ration (TMR) diets consisting of untreated rice stubble (URS), urea-treated rice stubble (UTRS), and urea-fungi-treated rice stubble (UFTRS).

Item	TMR Diet			SEM	*p* Value		
	URS	UTRS	UFTRS		Diet	Time	Diet × Time
Ruminal pH	6.60	6.55	6.54	0.04	0.549	<0.001	0.073
NH_3_-N concentration, mg/dL	15.88 ^b^	16.50 ^b^	17.87 ^a^	0.27	0.001	<0.001	0.324
Total VFA, mmol/L	88.93	88.50	83.54	4.12	0.599	0.001	0.327
VFA profile, mol/100 mol							
Acetic acid (C2)	58.93	59.87	61.78	0.80	0.066	0.053	0.220
Propionic acid (C3)	28.98	27.92	27.40	0.63	0.227	0.060	0.513
Butyric acid	12.09	12.21	10.82	0.49	0.118	0.032	0.189
C2:C3 ratio	2.09	2.20	2.32	0.08	0.141	0.081	0.295

^a,b^ Means in the same row with a different superscript, differ significantly (*p* < 0.05). NH_3_-N: ammonia. VFA: volatile fatty acid. SEM: standard error of mean

## Data Availability

The data presented in this study are available on request from the corresponding author.

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
