# Peer review of "Treatment of Rice Stubble with Pleurotus ostreatus and Urea Improves the Growth Performance in Slow-Growing Goats"

_animals, 2021, doi:10.3390/ani11041053_

Round 1

Reviewer 1 Report

Look at the attach file

Author Response

Please have a look at the attach file. Best regards.

Reviewer 2 Report

General comment:

The aim of the paper is of interest and falls within the scope of the Journal. In my opinion, the introduction section is well written and complete as well as the discussion section. However, some information in the material and methods section is lacking.

Please also see the specific comments.

Specific comments

Rows 72-74: Please clarify this sentence.

Table 2: You stated that the diets were isonitrogenous, what about the energy content of diets, were they isoenergetic? The energy content of diets could have influenced the in vivo performances.

Rows 118-119: The number of animals per experimental group (6) was rather low and I can guess that the power of the statistical test can be low. However, I can understand the difficulty to collect some samples (rows 151-152), but, if the power of the statistical test is low, the author should discuss the results and write the discussion section taking into this point.

Statistical analysis section: Since the number of animals per group was rather low, report if (and with which test) the assumptions of the anova were respected (e.g. normality of data distribution, homogeneity of variances,...)

Row 188: I do not agree, the initial body weight was not similar between experimental groups.

Rows 130-131: How did you assess the growth rate, considering only the initial and final weights or the weights obtained from the regression that involved all the weights that you have available per animals (problably, the second approach should be preferred)?

Rows 189-190: Why did you not assigned the animals to the experimental groups to have similar average initial live weight between experimental groups? Since the initial body weight can influence your variables, did you try to consider it in the statistical model (maybe as covariate …)? Please report other important information about animals (e.g. initial body condition, age,…)

Table 4: I have not completely clear how did you assess the digestibility.

Table 5: I think that the model you adopted for assessing the differences in the general mean values are not appropriate. Maybe a repeated measures model should be considered (if I understood well, each animal was assessed more times, at 0h, 2h, 4h, in my opinion, you can compare the experimental groups within sampling hour (0,2,4), but the same model can not be used to compare the general means).

Row 257: Please improve the discussion about the in vivo performance of animals.

Row 301: “the latter result of Zhang’s observation”, please clarify this sentence.

Author Response

(The authors gave the same response as above.)

Round 2

Reviewer 2 Report

In my opinion, the revised manuscript is improved. However, I have other things to highlight:

Rows 191-192: “the treatment means adjusted for the effect of the covariate.” Please report clearly if you considered the inclusion of covariance for all the variables or not, and which is the covariate. It seems that initial body weight was used as covariate only for growth performance. Did you test the assumptions for the analysis of covariance (linearity,…)?

Table 5: Since each animal was assessed more times (0h, 2h, 4h), I asked you to evaluate the possibility to considered a repeated measures model for the analysis of general means. With your statistical analysis for assessing the general means (anova), the anova assumption of the independence of samples is violated. Therefore, probably, if you are interested in assessing the differences between general means, in my opinion, you should use a model with the treatment effect (fixed factor) and time (repeated) with interaction. In this way, you can assessed the general means (treatment effect) and, if the interaction is significant, you can evaluate the differences between treatments within time (as you have already done).

Rows 335-336: I would suggest to report clearly that further studies with a larger sample size are needed.

Author Response

Please have a look at the attachment. Thank you.
